# The Association between Foot and Ulcer Microcirculation Measured with Laser Speckle Contrast Imaging and Healing of Diabetic Foot Ulcers

**DOI:** 10.3390/jcm10173844

**Published:** 2021-08-27

**Authors:** Onno A. Mennes, Jaap J. van Netten, Jeff G. van Baal, Riemer H. J. A. Slart, Wiendelt Steenbergen

**Affiliations:** 1Biomedical Photonic Imaging, University of Twente, 7522 NB Enschede, The Netherlands; r.h.j.a.slart@umcg.nl; 2Ziekenhuisgroep Twente, ZGT Academy, 7609 PP Almelo, The Netherlands; j.vbaal@zgt.nl; 3Amsterdam UMC, Department of Rehabilitation, Amsterdam Movement Sciences, University of Amsterdam, 1105 AZ Amsterdam, The Netherlands; j.j.vannetten@amsterdamumc.nl; 4Institute of Health and Biomedical Innovation, School of Clinical Sciences, Queensland University of Technology, Brisbane, QLD 4001, Australia; 5School of Medicine, Cardiff University, Cardiff CF 14 4XN, UK; 6Medical Imaging Center, Department of Nuclear Medicine and Molecular Imaging, University Medical Center Groningen, 9713 GZ Groningen, The Netherlands

**Keywords:** laser speckle contrast imaging, diabetes mellitus, diabetes complications, foot ulcer, microcirculation, peripheral artery disease, wound healing

## Abstract

Diagnosis of peripheral artery disease in people with diabetes and a foot ulcer using current non-invasive blood pressure measurements is challenging. Laser speckle contrast imaging (LSCI) is a promising non-invasive technique to measure cutaneous microcirculation. This study investigated the association between microcirculation (measured with both LSCI and non-invasive blood pressure measurement) and healing of diabetic foot ulcers 12 and 26 weeks after measurement. We included sixty-one patients with a diabetic foot ulcer in this prospective, single-center, observational cohort-study. LSCI scans of the foot, ulcer, and ulcer edge were conducted, during baseline and post-occlusion hyperemia. Non-invasive blood pressure measurement included arm, foot, and toe pressures and associated indices. Healing was defined as complete re-epithelialization and scored at 12 and 26 weeks. We found no significant difference between patients with healed or non-healed foot ulcers for both types of measurements (*p* = 0.135–0.989). ROC curves demonstrated moderate sensitivity (range of 0.636–0.971) and specificity (range of 0.464–0.889), for LSCI and non-invasive blood pressure measurements. Therefore, no association between diabetic foot ulcer healing and LSCI-measured microcirculation or non-invasive blood pressure measurements was found. The healing tendency of diabetic foot ulcers is difficult to predict based on single measurements using current blood pressure measurements or LSCI.

## 1. Introduction

Diabetes Mellitus is a metabolic disease and its patient population is growing worldwide, with a prevalence of 9.3% of the adults between 20 and 79 years old. A total of 463 million people are living with diabetes [1]. One of the major complications of diabetes is diabetic foot disease. Mortality, high morbidity, costs, and a reduced quality of life are all associated with diabetic foot disease [2,3,4,5]. Peripheral neuropathy and peripheral artery disease (PAD) are both major causes for diabetic foot ulceration, and PAD also contributes to poor healing outcomes [6,7]. Recognizing the levels of ischemia of the lower limb arteries is therefore essential in the treatment of foot ulcers in people with diabetes. However, this has been identified by various researchers and clinicians as one of the key challenges in diabetic foot disease [7,8]. 

Multiple methods are used to identify diabetic foot ulcers that are suspicious to poor healing as a result of PAD. Diagnostic arteriography and non-invasive blood pressure measurements are recommended in guidelines of the International Working Group on the Diabetic Foot (IWGDF) for the treatment of diabetic foot ulcers [9]. Indications for vascular consultation include ankle blood pressure < 50 mmHg, toe pressure < 30 mmHg, and ankle/brachial index (ABI) < 0.5 [6,10]. However, these non-invasive blood pressure measurements have various disadvantages. For example, it has been shown that ABI underestimates the prevalence of PAD in people with diabetes due to the arterial circular calcification of the media and the consequent non-compressibility [11,12]. Toe pressure does not reflect the vascular situation at the ulcer location, and does not measure microcirculatory status. Microcirculation can be estimated with transcutaneous oxygen pressure measurements (TcpO2), where a value above 25 mmHg has been demonstrated to predict ulcer healing potential; however, the presence of oedema can affect the accuracy of these measurements, and as this measurement is confined to the dorsal side of the foot it frequently does not reflect vascular status at the ulcer location [6,10,11,13]. In a recent systematic review on the performance of prognostic markers in the prediction of ulcer healing or amputation among foot ulcers in diabetes, it was concluded that wound healing was associated with a better perfused foot (skin perfusion pressure ≥ 40 mmHg, toe pressure ≥ 30 mmHg, or TcpO2 ≥ 25 mmHg) [8]. However, in most studies included in this systematic review, likelihood ratios of these tests in accurately predicting healing were small. While this is partly the result of coexisting factors (e.g., infection, comorbidities) also impacting on ulcer healing, it also suggests that other measurements overcoming the disadvantages of these non-invasive blood pressure measurements may result in better predictive values.

Novel optical imaging techniques such as laser speckle contrast imaging (LSCI) are available to complement the currently used non-invasive blood pressure measurements in people with diabetic foot disease [14]. LSCI is a non-invasive optical imaging technique able to measure blood flow in the skin [15,16]. In general, the reproducibility of LSCI is high, and it has low inter-subject variability [17,18,19,20]. LSCI is an interesting technique to measure blood flow in diabetic foot disease, because it has a widely validated track record of non-invasive in vivo blood flow measurements compared with other established methods of large-area microcirculation [14]. Furthermore, LSCI can provide non-invasive real-time feedback on changes in perfusion, and is able to monitor the microcirculation in the outer layer of the skin. Such microcirculation measurements give an indication of the perfusion directly in and around the ulcer, which overcomes the disadvantage of measurements such as ABI, toe pressure, and TcpO2. 

In a previous study among patients with a diabetic foot ulcer, we demonstrated that LSCI is a stable technique with a high inter- and intra-user reliability [21]. We concluded that LSCI can be used in the clinical setting complementing non-invasive blood pressure measurements. However, for assessing its clinical benefit, insight in its prognostic accuracy compared with non-invasive blood pressure measurements is required. Therefore, the aim of this study is to investigate and compare the prognostic values of LSCI and non-invasive blood pressure measurements in relation to healing of diabetic foot ulcers.

## 2. Materials and Methods

The clinical dataset used for this study was obtained as part of a larger study [21]. This study was approved by a registered medical ethics committee and registered in the Dutch trial register (NTR5116). A total of 33 patients with a diabetic foot ulcer participated and both the non-invasive blood pressure measurements and LSCI measurements of each ulcerated foot was available. This dataset was supplemented with people with a diabetic foot ulcer who were eligible for regular treatment, as LSCI was implemented in daily practice following completion of the above-mentioned study. All patients with a diabetic foot ulcer who were presenting at the outpatient clinic at ZGT Hospital, located in Almelo, the Netherlands, were scanned with LSCI. If the patient fulfilled the inclusion criteria and gave permission to use the data for scientific purposes, the patient data were included in this study. All examinations used in the current study were part of regular treatment and therefore the second cohort of this study was exempt from medical ethical review according to the Medical Research Involving Human Subjects Act in the Netherlands. These two cohorts together form the participants of this current single center, observational cohort study. All study actions were in line with the principles of the Declaration of Helsinki. 

Inclusion criteria were a confirmed diagnosis of type 1 or type 2 diabetes mellitus, and one foot ulcer (defined as break of the skin of the foot that involves at least the epidermis and part of the dermis [22]). Exclusion criteria were having multiple foot ulcers, an amputation of the forefoot or an amputation at a more proximal location of the foot (e.g., midfoot or hindfoot), moderate or severe foot infection (IWGDF grade 3 or 4; [23]), being incapacitated or undergoing cancer treatment. All patients were treated in accordance with the local protocol, which is based on the Dutch guidelines [24] and the IWGDF guidelines [25]. Treatment consisted of offloading, ulcer debridement and wound dressings, antibiotic treatment in case of mild infection, and blood pressure measurements to assess PAD, and surgical revascularization when required. Regular blood pressure measurements included both the non-invasive blood pressure measurements (i.e., arm pressure, ankle pressure, ABI, toe pressure, and TcpO2). Regular microcirculatory measurements included LSCI scans in and around the ulcer location. 

Measurements were performed after ulcer debridement, and consisted of first doing LSCI scans, followed by non-invasive blood pressure measurements. LSCI scans were performed of the ulcer foot with a PeriCam PSI NR (Perimed AB, Stockholm, Sweden). Either the plantar or the dorsal side of the foot was scanned, depending on the ulcer location, with the ulcer location to be included in the scan. Perfusion was expressed in perfusion units (PU). During the scan, the patient lay supine on the examination table barefoot. After 5 min, for the patient to get used to the room temperature (kept between an ambient 21–22 degrees), the LSCI scans were acquired. During the scans, three different time periods of interest (TOI) were measured: baseline, biological zero, and post-occlusion hyperemia measurements. The baseline was a measurement in the first stage of the scan when the measured perfusion was stable on visual inspection for 30 s. Subsequently, a cuff around the ankle was inflated to stop blood flow to the foot. During this time the perfusion dropped to the biological zero value of the patient. When the perfusion did not further decrease for 30 s, the biological zero was measured. After this measurement, the ankle cuff was released. The maximum measured blood flow after release of pressure was used as the post-occlusion hyperemia value. During each TOI, different regions of interest (ROI) of the foot were measured (i.e., foot, ulcer, and ulcer edge). As described in detail in our previous paper [21], each ROI was manually selected in the scans in order to measure the mean perfusion of different areas of the foot and ulcer. The ROIs were positioned at the beginning of each TOI and repositioned during the scan to correct for possible movement of the foot during the scan [21]. Non-invasive blood pressure measurements consisted of measuring arm pressure, ankle pressure, toe pressure, and TcpO2, with a PeriFlux 6000 (Perimed AB, Stockholm, Sweden), all according to the manufacturer’s instructions. 

Each patient was classified as non-ischemic, ischemic, or critical-ischemic based on non-invasive blood pressure measurements, following IWGDF criteria [6,26]. Patients were classified as critical-ischemic when ABI ≤ 0.39, or ankle pressure < 50 mmHg, or toe pressure or TcpO2 < 30 mmHg. Patients not classified as critical-ischemic but had an ABI between 0.4–0.79, or an ankle pressure between 50–100 mmHg, or a toe pressure or TcpO2 between 30–59 mmHg, were classified as ischemic. Patients were classified as non-ischemic with ABI ≥ 0.8 and an ankle pressure > 100 mmHg and a toe pressure and TcpO2 ≥ 60 mmHg [6,10,26].

Clinical background and different parameters (Table 1) of the patient were obtained at baseline. The level of neuropathy was measured with a 10 g Semmes–Weinstein monofilament [6], HbA1c was measured with blood tests, and other parameters such as smoking were collected or measured during anamnesis. Follow-up for outcomes was until ulcer healing or for a maximum 26 weeks. Healing of the foot ulcer was defined as complete re-epithelialization of the ulcer without revascularization or major amputation [22] and was scored by an experienced clinician at 12 and 26 weeks during the outpatient clinic visits. Patients who died, who underwent revascularization, or major amputation were excluded. 

### Statistical Analysis

To investigate differences between the healed and non-healed participants at 12 and 26 weeks, t-tests were conducted for all numerical variables and a Chi^2^ test for all categorical variables. Statistical relevance was considered with a *p*-value less than 0.05. ROC curves were created and the sensitivity and specificity of different parameters were calculated. The thresholds to calculate positive and negative likelihood ratio (LLR+ and LLR–) were chosen based on the highest combination of both sensitivity and specificity. A LLR– between 0.5–1 or LLR+ between 1–5 indicates no small change, while a LLR– between 0.1–0.5 or LLR+ between 5–10 were considered as moderate. LLR– below 0.1 or LLR+ above 10 were considered as large effect [8].

## 3. Results

A total of 61 patients were included. One patient died during follow up, two patients underwent major amputation, and five patients underwent revascularization. Of the 53 patients included for analysis, 23 (43.4%) healed within 12 weeks, 36 (67.9%) in 26 weeks, while 17 patients (32.1%) did not heal in 26 weeks or received revascularization treatment (Figure 1, Table 1). 

Patients were on average 67 years, predominantly male, and with an average BMI of 29.7 (Table 1). Average healing percentages for the non-ischemic, ischemic, and critical-ischemic groups were 4.3%, 69.6%, and 26.1% at 12 weeks and 11.1%, 63.9%, and 25.0% at 26 weeks. There were no significant differences between healed and non-healed patients at 12 weeks (*p* = 0.925) and also no significant differences between these groups at 26 weeks (*p* = 0.275; Table 1). Furthermore, for the majority of other patient characteristics, no significant difference was found between healers and non-healers (*p*-values ranging from 0.027–0.949; Table 1). 

There were no significant differences in any of the perfusion measurements between the healed and non-healed group, neither for LSCI at the foot, ulcer, or ulcer edge, nor for any of the non-invasive blood pressure measurements (*p*-values ranging from 0.136–0.983; Table 2). There were also no significant differences when we compare the 95% confidence intervals of both the LSCI perfusion measurements and non-invasive blood pressure measurements (Figure 2). 

The ROC curves for LSCI scans and non-invasive blood pressure measurements demonstrated poor to moderate sensitivity and specificity (Figure 3).

Both LLR+ and LLR– showed a small to no effect (LLR+ 1.06–4.72; LLR– 0.36–0.89; Figure 3, Table 3). The largest effect for prognosis of healing at 12 weeks was found for LSCI at the ulcer during baseline or post-occlusive peak (LLR–: 0.36). The largest effect for prognosis of healing at 26 weeks was found for LSCI at the ulcer edge during baseline or post-occlusive peak (LLR+: 4.72 and 2.60) and for ankle and toe pressure (LLR–: 0.40).

With no significant differences found between patients who healed and those who did not heal, we repeated all tests for the group of participants classified as ischemic only (*n* = 28). We chose to do so, because advanced blood pressure assessment is most important in this group from a clinical perspective, as these are patients for whom diagnosis and prognosis are in a grey area. However, these post hoc analyses did not result in different findings (results not shown); again, no differences were seen in blood pressure measurements between patients who healed and patients who did not heal.

## 4. Discussion

The aim of this study was to investigate the association between foot and ulcer (micro) circulation (measured with both LSCI and non-invasive blood pressure measurements) and healing of diabetic foot ulcers at 12 and 26 weeks. We found no significant differences in any of the measurements between the group of healed and non-healed patients, neither at 12 nor 26 weeks. Positive and negative likelihood ratios showed no or only small effects. In our cohort, both LSCI and non-invasive blood pressure measurements were not useful as a standalone prognostic test for diabetic foot ulcer healing. This result is not in line with the outcomes of a recent systematic review by Forsythe et al. [8] in which it was concluded that some non-invasive blood pressure measurements may have prognostic value. However, the majority of studies included in this review showed similar likelihood ratios as found in the current study [8]. This implies that prognostic quality of non-invasive blood pressure measurements on its own are not always a valuable predictor for healing of diabetic foot ulcers. 

While our study had a different approach in calculating the cut-off values for different non-invasive bedside blood pressure measurement tests, we can still use the results of the studies included in Forsythe et al. [8] to put the effect of the found likelihood ratios in perspective. First, in our study, the cut-off values for the different blood pressure measurements were based on the optimal combination of both sensitivity and specificity for ulcer healing. This is a different approach compared with other studies in which they used fixed cut-off values and in which they calculated corresponding likelihood ratios based on those values. We chose this approach, because no cut-off values for LSCI measurements are available yet. Therefore, it was necessary to find the cut-off values with the highest prognostic power. To compare LSCI with the non-invasive blood pressure measurements, we used the same technique and calculations with this bedside test as well. Despite this difference in approach, the likelihood ratios were comparable. For example, we found a LLR+ for healing after 12 and 26 weeks based on the ankle pressure of 1.53 and 1.46 with a cut-off values of >153mmHg and >96.0 mmHg. Other studies found an LLR+ of 1.08 (>50 mmHg) [28], 1.46 (≥50 mmHg) [29], 2.52 (≥80 mmHg) [29], 3.24 (≥70 mmHg) [30], and 6.40 (≥100 mmHg) [31]. Although the studies with a higher threshold (>70 mmHg) showed a higher LLR+, this effect was still small (LLR+: 2.52–3.24) to moderate (LLR+ 6.40), while other studies observed no change in effect based on ankle pressure measurements (LLR+: 1.08–1.46). 

Similar findings are seen when comparing the found LLR+ and LLR– for toe pressure and TcpO2. Our study found LLR+ and LLR– for toe pressure and TcpO2 of 1.46, 0.40, and 1.24, 0.47, respectively. Other studies reported similar LLR+ and LLR– for toe pressure measurements. For example 1.12, 0.88 [28]; 1.28, 0.33 [29]; 2.47, 0.21 [30]; 2.88, 0.64 [32]; 4.30, 0.25 [29]; and 5.00, 0.88 [32]. Although the LLR are not exactly identical, the results are similar in effect and range from no effect (LLR+: 1–2; LLR–: 0.5–1), to a small effect (LLR+: 2–5; LLR–: 0.2–0.5), comparable to our findings of no effect (LLR+: 1.46) and a small effect (LLR–: 0.40). 

When we compared our LLR+ for TcpO2 with other LLR+ values, some studies did find larger effects: LLR+ of 10.03 [32] and 5.14 [33] were found for TcpO2 thresholds ≥30 mmHg, indicating a moderate to large prognostic effect. However, those findings were not unanimous as other studies found lower LLR+ (1.21 and 2.73) [34,35]. This is an indication that the prognostic power of different tests are influenced by the specific patient populations and other factors such as environment and time period.

As this is the first study to investigate LLR+ and LLR– for LSCI in relation to diabetic foot ulcer healing, there are no findings for direct comparison. However, in light of the above-mentioned studies, our findings are within the expected range. Despite the advantages of measuring at and around the exact ulcer location, and including both baseline perfusion values and stress-test values, LSCI did not result in improved prognostic likelihood ratios for ulcer healing in this cohort, compared with regular non-invasive blood pressure measurements.

The following limitations of this study should be considered. First, the combination of more than one prognostic test may provide more useful information on the probability of healing than a single test or test used in isolation [8]. However, in the current study, we analyzed the different blood pressure tests individually instead of combining them. While it is interesting to do so in a follow-up study, with the low likelihood ratios found, the benefits of combining may be small. 

Second, the exclusion of patients that underwent a major amputation can be considered as another limitation. A major amputation could have been considered as endpoint, too, in addition to wound healing. This could be useful for clinicians in order to identify the patients with a higher probability of healing without revascularization to pursue a conservative approach. Furthermore, it could be of importance to identify patients with an unacceptable high risk of a major amputation. For those patients, adequate revascularization should be a priority [8]. Therefore, amputation incidence can help in assessing the impact of disease. However, it is not necessarily a good measure of the quality of care and amputation incidence is partially based on the clinical choice of the attending physician [36]. Therefore, we decided to focus on the healing of the diabetic foot as a biological endpoint. 

Third, the follow-up period of six months and the use of two measurement moments (at 12 and 26 weeks) dichotomizes healing, rather than using the more detailed time to healing in days or weeks. This dichotomization results in diverse groups, where both short healing times (<4 weeks) and long healing times (>20 weeks) could end up in the same group. In further research it might be better to use time to heal (in weeks) as an endpoint for that study. This provides a better understanding and more useful parameter to obtain meaningful insights into the patients’ healing tendency, since the time needed to heal a chronic diabetic foot ulcers usually varies a lot. We decided to use a different approach in this study for several reasons. First, we wanted to compare our results with previous research [8]. Second, dichotomization is recommended for the assessment of data in diabetic foot research (e.g., [37]). Third, diagnostic values cannot be calculated for a continuous outcome measure. 

A fourth limitation of this study is that drug use and specific additional treatment of the patient (for example offloading or wound dressings) were not taken into account in this study. Where the first might influence microcirculation measurements, the latter might influence healing outcomes. However, because all patients were treated in the same center and by the same clinicians, clinical decisions were considered similar, and therefore not accounted for in analyses. Furthermore, while some drugs might affect microcirculation, no previous study on prognosis has found an effect of such drugs on likelihood ratios for prognosis [8]. 

Finally, it is questionable whether we can compare our results with outcomes of previous studies. Although we see comparable results, it is likely that the included populations differ. Whereas in the past, the majority of patients with diabetic foot ulcers had been treated in hospitals, currently only the more complex cases visit hospitals or specialized care centers for diabetic foot ulcers. For future research it would be interesting to compare and validate our findings with more recent studies. 

## 5. Conclusions

No association between healing of diabetic foot ulcers and microcirculation measured with LSCI or non-invasive blood pressure measurements was found. We can conclude that both types of measurements were not useful as a standalone prognostic instrument for diabetic foot ulcer healing.

## Figures and Tables

**Figure 1 jcm-10-03844-f001:**
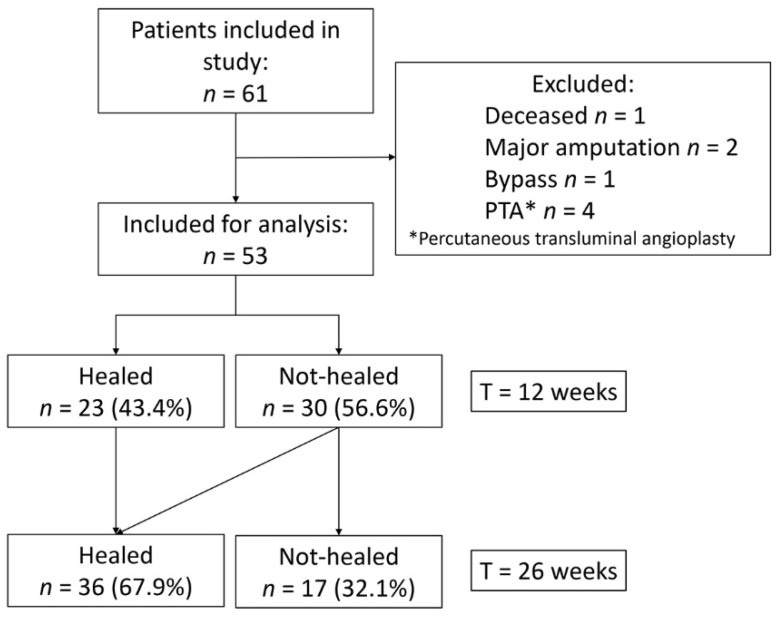
Schematic overview of patient population and clinical outcomes at 12 and 26 weeks.

**Figure 2 jcm-10-03844-f002:**
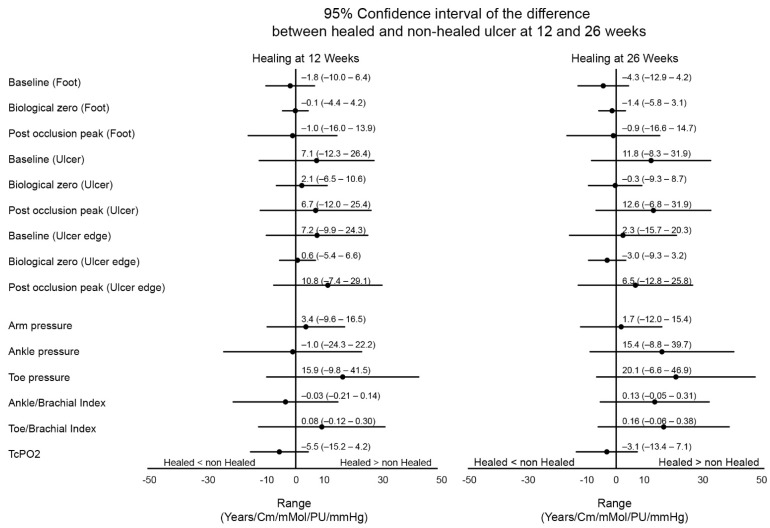
Difference in range of parameters between healed and non-healed patients at 12 and 26 weeks.

**Figure 3 jcm-10-03844-f003:**
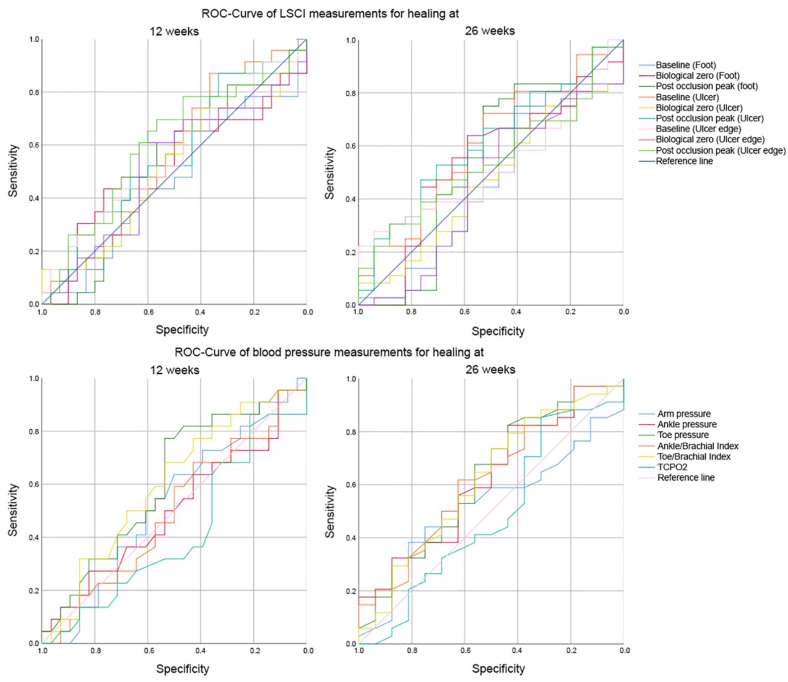
ROC curves showing sensitivity and specificity for laser speckle contrast imaging measurements and non-invasive blood pressure measurements as prognostic tests for ulcer healing at 12 or 26 weeks.

**Table 1 jcm-10-03844-t001:** Patient characteristics at baseline, and separated between healed and non-healed patients at 12 and 26 weeks.

Variable		Baseline	12 Weeks			26 Weeks		
		Mean ± SD N (%)	Healed Mean ± SD	Non-Healed Mean ± SD	*p*-Value	Healed Mean ± SD	Non-Healed Mean ± SD	*p*-Value
**Patient Characteristics**		53 (100%)	23 (43.4%)	30 (56.6%)		36 (67.9%)	17 (32.1%)	
**Age (Years)**		66.7 ± 12.8	68.9 ± 13.1	65.1 ± 12.7	0.300	67.3 ± 11.9	65.7 ± 15.1	0.679
**Gender**	Male	42 (79.2%)	18 (78.3%)	24 (80%)	0.877	30 (83.3%)	12 (70.6%)	0.286
	Female	11 (20.8%)	5 (21.7%)	6 (20%)		6 (16.7%)	5 (29.4%)	
**Height (cm)**		179.4 ± 9.6	179.1 ± 11.1	179.7 ± 8.4	0.858	179.7 ± 10.7	178.7 ± 6.6	0.744
**Weight (kg)**		96.0 ± 19.9	98.0 ± 21.4	94.4 ± 18.8	0.546	96.9 ± 20.1	93.9 ± 20.0	0.660
**BMI**		29.7 ± 5.5	30.4 ± 5.5	29.2 ± 5.6	0.475	29.9 ± 5.2	29.4 ± 6.4	0.808
**HbA1c (mmol/mol)**		63.6 ± 21.0	59.5 ± 14.4	67.6 ± 25.6	0.221	63.0 ± 15.9	65.7 ± 33.4	0.729
**Smoking**	Yes	11 (52.4%)	3 (42.9%)	8 (57.1%)	0.590	6 (46.2%)	5 (62.5%)	0.436
	No	4 (19.0%)	1 (14.3%)	3 (21.4%)		2 (16.4%)	2 (25.0%)	
	Stopped	6 (28.6%)	3 (42.9%)	3 (21.4%)		5 (38.5%)	1 (12.5%)	
	Unknown	32	16	16		23	9	
**Diabetes Type**	1	3 (5.7%)	1 (4.3%)	2 (6.7%)	0.717	2 (5.6%)	1 (5.9%)	0.962
	2	50 (94.3%)	22 (95.7%)	28 (90.0%)		34 (94.4%)	16 (88.2%)	
**Diabetes** **Duration**	≤10 years	20 (43.5%)	8 (38.1%)	12 (48.0%)	0.394	12 (36.4%)	8 (61.5%)	0.180
	>10 years	26 (56.5%)	13 (61.9%)	13 (52.0%)		21 (63.6%)	5 (38.5%)	
	Unknown	7	2	5		3	4	
**Dialysis**	Yes	3 (5.7%)	1 (4.3%)	2 (6.7%)	0.266	2 (5.6%)	1 (5.9%)	0.998
	No	47 (88.7%)	22 (95.7%)	25 (83.3%)		32 (88.9%)	15 (88.2%)	
	In the past	3 (5.7%)	0 (0.0%)	3 (10.0%)		2 (5.6%)	1 (5.9%)	
**Infections**	Yes	8 (15.1%)	4 (17.4%)	4 (13.3%)	0.683	5 (13.9%)	3 (17.6%)	0.721
	No	45 (84.9%)	19 (82.6%)	26 (86.7%)		31 (86.1%)	14 (82.4%)	
**Neuropathy**	Yes	48 (96.0%)	22 (100.0%)	26 (92.9%)	0.201	35 (100.0%)	13 (86.7%)	0.027 *
	No	2 (4.0%)	0 (0.0%)	2 (7.1%)		0 (0.0%)	2 (13.3%)	
	Unknown	3	1	2		1	2	
**UT-classification**					0.776			0.704
	0A	4 (7.5%)	2 (8.7%)	2 (6.7%)		3 (8.3%)	1 (5.9%)	
	1A	30 (56.6%)	15 (65.2%)	15 (50.0%)		22 (62.9%)	8 (47.1%)	
	1B	1 (1.9%)	0 (0.0%)	1 (3.3%)		0 (0.0%)	1 (5.9%)	
	1C	1 (1.9%)	0 (0.0%)	1 (3.3%)		1 (2.8%)	0 (0.0%)	
	2A	5 (9.4%)	1 (4.3%)	4 (13.3%)		3 (8.3%)	2 (11.8%)	
	2B	5 (9.4%)	2 (8.7%)	3 (10.0%)		3 (8.3%)	2 (11.8%)	
	3A	3 (5.7%)	1 (4.3%)	2 (6.7%)		2 (5.6%)	1 (5.9%)	
	3B	3 (5.7%)	2 (8.7%)	1 (3.3%)		2 (5.6%)	1 (5.9%)	
	3C	1 (1.9%)	0 (0.0%)	1 (3.3%)		0 (0.0%)	1 (5.9%)	
**History of Ulcers**	Yes	31 (58.5%)	15 (65.2%)	16 (53.3%)	0.384	20 (55.6%)	11 (64.7%)	0.528
	No	22 (41.5%)	8 (34.8%)	14 (46.7%)		16 (44.4%)	6 (35.3%)	
**Minor** **Amputation**	Yes	8 (15.1%)	2 (8.7%)	6 (20.0%)	0.255	5 (13.9%)	3 (17.6%)	0.721
	No	45 (84.9%)	21 (91.3%)	24 (80.0%)		31 (86.1%)	14 (82.4%)	
**Vascular** **Status**	Non- ischemic	7 (13.2%)	1 (4.3%)	6 (20.0%)	0.925	4 (11.1%)	3 (17.6%)	0.275
	Ischemic	28 (52.8%)	16 (69.6%)	12 (40.0%)		23 (63.9%)	5 (29.4%)	
	Critical- ischemic	18 (34.0%)	6 (26.1%)	12 (40.0%)		9 (25.0%)	9 (52.9%)	

* *p* < 0.05; note: UT-classification is the University of Texas Diabetic Wound Classification [27].

**Table 2 jcm-10-03844-t002:** Mean values of laser speckle contrast imaging (in perfusion units (PU)) and non-invasive blood pressure measurements (mmHg) at baseline and between patients with healed versus non-healed foot ulcers.

Variable		Baseline		12 Weeks			26 Weeks	
		Mean ± SD N (%)	Healed Mean ± SD	Non-Healed Mean ± SD	*p*-Value	Healed Mean ± SD	Non-Healed Mean ± SD	*p*-Value
**Laser Speckle Contrast Imaging (PU)**								
**Foot**								
Baseline		50.3 ± 14.6	49.3 ± 15.1	51.1 ± 14.5	0.654	49.4 ± 13.9	52.3 ± 16.3	0.508
Biological zero		12.8 ± 7.7	12.7 ± 7.3	12.8 ± 8.1	0.959	12.5 ± 6.5	13.5 ± 10	0.637
Post occlusion peak		77.3 ± 26.6	76.7 ± 24.4	77.8 ± 28.6	0.889	77.4 ± 23.2	77.2 ± 33.6	0.983
**Ulcer**								
Baseline		104.8 ± 34.6	108.8 ± 33	101.8 ± 36.1	0.467	109.1 ± 35.7	95.8 ± 31.2	0.197
Biological zero		25.2 ± 15.3	26.4 ± 17.9	24.3 ± 13.2	0.631	25 ± 16.3	25.7 ± 13.4	0.884
Post occlusion peak		104.0 ± 33.4	107.8 ± 32.6	101.1 ± 34.3	0.473	108.2 ± 35.2	95.2 ± 28.2	0.190
**Ulcer Edge**								
Baseline		92.2 ± 30.7	96.3 ± 33.4	89.1 ± 28.6	0.402	94.2 ± 33.8	88.1 ± 23	0.509
Biological zero		20.1 ± 10.7	20.5 ± 10.8	19.8 ± 10.9	0.840	19.4 ± 10	21.7 ± 12.3	0.465
Post occlusion peak		102.0 ± 32.9	108.1 ± 33.9	97.3 ± 31.9	0.239	104.8 ± 35.3	96 ± 27.4	0.373
**Non-invasive Blood Pressure Measurements (mmHg)**								
Ankle		121.9 ± 41.0	121.3 ± 46.3	122.3 ± 37.2	0.931	126.9 ± 41.4	110.4 ± 39	0.183
Toe		88.7 ± 45.3	97.7 ± 45.1	81.8 ± 45.1	0.220	95.4 ± 45.2	75.2 ± 43.8	0.136
ABI		0.90 ± 0.31	0.88 ± 0.3	0.92 ± 0.32	0.698	0.94 ± 0.32	0.82 ± 0.3	0.188
TBI		0.68 ± 0.37	0.73 ± 0.32	0.64 ± 0.41	0.410	0.73 ± 0.38	0.57 ± 0.34	0.151
Tc*p*O_2_		47.9 ± 17.5	44.8 ± 15.1	50.3 ± 19.1	0.262	46.8 ± 14.1	50.1 ± 23.6	0.526

Note: PU = perfusion units; ABI = ankle/brachial index; TBI = toe/brachial index; and Tc*p*O_2_ = transcutaneous oxygen pressure.

**Table 3 jcm-10-03844-t003:** Threshold, sensitivity, and specificity for non-invasive blood pressure measurements and laser speckle contrast imaging measurements, for healing after 12 weeks and 26 weeks.

**12 Weeks**	**Threshold**	**AUC**	**Sensitivity**	**Specificity**	**LLR+**	**LLR** **–**
**Laser speckle contrast imaging (PU)**						
**Foot**						
Baseline	43.5 PU	0.467	0.696	0.400	1.16	0.76
Biological zero	14.3 PU	0.528	0.435	0.767	1.86	0.74
Post occlusion peak	73.5 PU	0.517	0.609	0.567	1.40	0.69
**Ulcer**						
Baseline	84.3 PU	0.558	0.870	0.367	1.37	0.36 **
Biological zero	15.7 PU	0.510	0.739	0.400	1.23	0.65
Post occlusion peak	89.4 PU	0.561	0.870	0.333	1.30	0.39 **
**Ulcer edge**						
Baseline	103.0 PU	0.552	0.435	0.700	1.45	0.81
Biological zero	19.9 PU	0.519	0.609	0.633	1.66	0.62
Post occlusion peak	96.6 PU	0.603	0.696	0.567	1.61	0.54
**Non-invasive blood pressure measurements (mmHg)**						
Arm pressure	130.5 mmHg	0.528	0.636	0.500	1.27	0.73
Ankle pressure	153.0 mmHg	0.500	0.273	0.821	1.53	0.89
Toe pressure	77.5 mmHg	0.608	0.773	0.536	1.66	0.42 **
Ankle brachial index	0.83	0.494	0.682	0.429	1.19	0.74
Toe brachial index	0.57	0.599	0.682	0.536	1.47	0.59
Tc*p*O_2_	30.5 mmHg	0.416	0.818	0.214	1.04	0.85
**26 weeks**	**Threshold**		**Sensitivity**	**Specificity**	**LLR+**	**LLR** **–**
**Laser speckle contrast imaging (PU)**						
**Foot**						
Baseline	41.9 PU	0.454	0.722	0.412	1.23	0.67
Biological zero	10.9 PU	0.540	0.639	0.588	1.55	0.61
Post occlusion peak	62.3 PU	0.541	0.750	0.529	1.59	0.47 *
**Ulcer**						
Baseline	92.3 PU	0.606	0.722	0.529	1.53	0.52
Biological zero	12.7 PU	0.469	0.750	0.294	1.06	0.85
Post occlusion peak	109.5 PU	0.609	0.472	0.765	2.01 *	0.69
**Ulcer edge**						
Baseline	118.5 PU	0.525	0.278	0.941	4.72 *	0.77
Biological zero	14.0 PU	0.455	0.667	0.471	1.26	0.71
Post occlusion peak	123.0 PU	0.547	0.306	0.882	2.60 *	0.79
**Non-invasive blood pressure measurements (mmHg)**						
Ankle pressure	96.0 mmHg	0.619	0.824	0.438	1.46	0.40 *
Toe pressure	54.0 mmHg	0.626	0.824	0.438	1.46	0.40 *
Ankle brachial index	0.89	0.619	0.618	0.625	1.65	0.61
Toe brachial index	0.51	0.618	0.735	0.500	1.47	0.53
Tc*p*O_2_	30.5 mmHg	0.484	0.853	0.313	1.24	0.47 *

Note: AUC = area under the curve; LLR+ = positive likelihood ratio; LLR– = negative likelihood ratio; Tc*p*O_2_= transcutaneous oxygen pressure measurements; * small effect, ** moderate effect.

## Data Availability

The data used to support the findings of this study are included within the article. Additional source data can be requested from the corresponding authors (O.A.M. and W.S.).

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
