# Peer review of "The Association between Foot and Ulcer Microcirculation Measured with Laser Speckle Contrast Imaging and Healing of Diabetic Foot Ulcers"

_jcm, 2021, doi:10.3390/jcm10173844_

Round 1
Reviewer 1 Report
In this manuscript, researched about The Association between Foot and Ulcer Microcirculation 2 measured with Laser Speckle Contrast Imaging and Healing of Diabetic Foot Ulcers
However the research could be improved, even though has an important clinical message, and should be of interest to the JCM readers.
This research showed The Association between Foot and Ulcer Microcirculation 2 measured with Laser Speckle Contrast Imaging and Healing
From my humble point of view this research was well-organized and show precise results.
However,I recommend review carefully and accurately this manuscript after major revision because of the following issues.
1.) Title:
The title should say something to impress readers. Please perform a new title, which gives clinical relevance of the paper.
2.) Introduction
This section shows clear structure with progression on importance of treatment of diabetic foot wound, beside is a condition frequently described in the literature. Within these sections it was highlighted that prevalence in the population. This led nicely to the purpose of the study.
Them authors need discuss this later on in the discussion portion and highlight the findings in the conclusion and abstract.
Introduction may be improved adding new information in order to provide an adequate state-of-the-art including some references. I suggest to include this reference to complete this requirement related to Diabetic foot complications that authors do not included - Gijón-Noguerón et al doi: 10.1177/1938640015585963
3 .) Materials and methods
All methods are supported according to adequate methodology how was conducted the study.
4.) Results
The results section is enough appropriate according to the developed methods and the journal´s scope.
6.) Discussion.
This section needs to be improves in order to understanding of the results section comparing with novel and adequate studies. I would suggest to include information related to laser for example authors should discuss their result with the achivements of the research of Camacho Perea et al to complete the trend of research nowadays related to laser heal plantar ulcers
DOI: 10.5209/RICP.62340
On the other hand, discussion section include future research studies secondary to the current findings of this study. However authors should discuss their achivement with regard to another metabolic parameters such as salivary amylase in the case of diabetes
I suggest to include the following reference to complete this requeriment
Pérez-Ros P, et al. doi: 10.3390/diagnostics11030453.
Reviewer 2 Report
I reviewed the manuscript “The Association between Foot and Ulcer Microcirculation 2 measured with Laser Speckle Contrast Imaging and Healing of 3 Diabetic Foot Ulcers“ where the authors looked for an association between microsimulation and healing of diabetic foot ulcers (DFU).
They found no significant difference between helped and unhealed patients regarding different measurements of the microsimulation in a foot. I think it is always important to report no-effect and thank authors for their manuscript.
However, I have some questions/comments
1. Why one of the exclusion criteria was a more proximal location of an ulcer? And what is it mean “more”? How was it decided if an ulcer was proximal enough?
2. What was the definition of healing there? Only a complete re-epithelialisation? Some clinicians defined healing as macroscopically complete epithelialisation that had to be reconfirmed after up to six months. Did the author also assume some time gap between epithelialisation and healing?
3. I think that using a t-test without testing for normality is inappropriate. I would suggest using non-parametric tests of medians comparison.
4. Also, exclusion of some patients who died, underwent revascularisation or had major amputations might lead to losing the important information about the process because these events are comparing with the primary healing. I think that competing event analysis (Cox regression, for example, or Kaplan-Meyer curves) would fit better into the process.
5. The authors discussed some “general” differences in the results between groups while all results are non-significant. I would skip these speculations.
6. Table 3: it’s not clear where Laser Speckle Contrasting Imaging is in the table.
Reviewer 3 Report
Dear authors,
Once I have analyzed your article, I indicate my conclusions:
1. The study presents the results of an original investigation.
2. The reported results have not been published elsewhere.
3. The experiments, statistics and other analyzes are carried out at a high technical level and are described in sufficient detail.
4. Conclusions are appropriately presented and supported by data.
5. The article is presented in an intelligible manner and is written in standard English.
6. The research meets all applicable standards for the ethics of experimentation and the integrity of research.
7. The article adheres to appropriate community standards and reporting guidelines for data availability.
8. I recommend inserting bibliography on the prevention of diabetic foot such as:
Astasio-Picado Á, Escamilla Martínez E, Gómez-Martín B. Comparative thermal map of the foot between patients with and without diabetes through the use of infrared thermography. Enferm Clin (Engl Ed). 2020 Mar-Apr;30(2):119-123. English, Spanish. doi: 10.1016/j.enfcli.2018.11.002. Epub 2019 Jan 7. PMID: 30630674.
Sandi S, Yusuf S, Kaelan C, Mukhtar M. Evaluation risk of diabetic foot ulcers (DFUs) using infrared thermography based on mobile phone as advanced risk assessment tool in the community setting: A multisite cross-sectional study. Enferm Clin. 2020 Mar;30 Suppl 2:453-457. English, Spanish. doi: 10.1016/j.enfcli.2019.07.136. PMID: 32204210.
Astasio-Picado Á, Martínez EE, Gómez-Martín B. Comparison of Thermal Foot Maps between Diabetic Patients with Neuropathic, Vascular, Neurovascular, and No Complications. Curr Diabetes Rev. 2019;15(6):503-509. doi: 10.2174/1573399815666190206160711. PMID: 30727903.
Round 2
Reviewer 1 Report
After to read carefully the reviewed version I recomend to reject this manuscript due to the fact that the authors do not attended the reviewer recomendations
In fact, they only have adressed a small part of the reviewers comment
And in the case of the references and state-of-art it is necessary to include the following references to increase the level evidence to make interesting to JCM readers
Pérez-Ros P, et al. doi: 10.3390/diagnostics11030453
Camacho Perea et al DOI: 10.5209/RICP.62340
Gijón-Noguerón et al doi: 10.1177/1938640015585963